# BISPECTRAL NEURAL NETWORKS

**Sophia Sanborn**[1,2]
sanborn@berkeley.edu

**Christian Shewmake**[1,2]
shewmake@berkeley.edu

**Bruno Olshausen**[1,2]
baolshausen@berkeley.edu

**Christopher Hillar**[3]
hillarmath@gmail.com

[1]Redwood Center for Theoretical Neuroscience
[2]University of California, Berkeley
[3]Awecom, Inc.

## ABSTRACT

We present a neural network architecture, Bispectral Neural Networks (BNNs) for learning representations that are invariant to the actions of compact commutative groups on the space over which a signal is defined. The model incorporates the ansatz of the bispectrum, an analytically defined group invariant that is *complete*—that is, it preserves all signal structure while removing only the variation due to group actions. Here, we demonstrate that BNNs are able to simultaneously learn groups, their irreducible representations, and corresponding equivariant and complete-invariant maps purely from the symmetries implicit in data. Further, we demonstrate that the completeness property endows these networks with strong invariance-based adversarial robustness. This work establishes Bispectral Neural Networks as a powerful computational primitive for robust invariant representation learning.

## 1 INTRODUCTION

A fundamental problem of intelligence is to model the transformation structure of the natural world. In the context of vision, translation, rotation, and scaling define symmetries of object categorization—the transformations that leave perceived object identity invariant. In audition, pitch and timbre define symmetries of speech recognition. Biological neural systems have learned these symmetries from the statistics of the natural world—either through evolution or accumulated experience. Here, we tackle the problem of learning symmetries in artificial neural networks.

At the heart of the challenge lie two requirements that are frequently in tension: *invariance* to transformation structure and *selectivity* to pattern structure. In deep networks, operations such as `max` or `average` are commonly employed to achieve invariance to local transformations. Such operations are invariant to many natural transformations; however, they are also invariant to unnatural transformations that destroy image structure, such as pixel permutations. This lack of selectivity may contribute to failure modes such as susceptibility to adversarial perturbations [1], excessive invariance [2], and selectivity for textures rather than objects [3]. Thus, there is a need for computational primitives that selectively parameterize natural transformations and facilitate robust invariant representation learning.

An ideal invariant would be *complete*. A complete invariant preserves all pattern information and is invariant only to specific transformations relevant to a task. Transformations in many datasets arise from the geometric structure of the natural world. The mathematics of *groups* and their associated objects give us the machinery to precisely define, represent, and parameterize these transformations, and hence the problem of invariant representation learning. In this work, we present a novel neural network primitive based on the *bispectrum*, a complete invariant map rooted in harmonic analysis and group representation theory [4].

Bispectral Neural Networks flexibly parameterize the bispectrum for arbitrary compact commutative groups, enabling both the group and the invariant map to be learned from data. The architecture is remarkably simple. It is comprised of two layers: a single learnable linear layer, followed by a fixed

collection of triple products computed from the output of the previous layer. BNNs are trained with an objective function consisting of two terms: one that collapses all transformations of a pattern to a single point in the output (invariance), and another that prevents information collapse in the first layer (selectivity).

We demonstrate that BNNs trained to separate orbit classes in augmented data learn the group its Fourier transform, and corresponding bispectrum purely from the symmetries implicit in the data (Section 4.1). Because the model has learned the fundamental structure of the group, we show that it generalizes to novel, out-of-distribution classes with the same group structure and facilitates downstream group-invariant classification (Section 4.2). Further, we demonstrate that the trained network inherits the completeness of the analytical model, which endows the network with strong adversarial robustness (Section 4.3). Finally, we demonstrate that the weights of the network can be used to recover the group Cayley table—the fundamental signature of a group's structure (Section 4.4). Thus, an explicit model of the group can be learned and extracted from the network weights. To our knowledge, our work is the first to demonstrate that either a bispectrum or a group Cayley table can be learned from data alone. Our results set the foundation of a new computational primitive for robust and interpretable representation learning.

## 1.1 RELATED WORK

The great success and efficiency of convolutional neural networks owe much to built-in *equivariance* to the group of 2D translations. In recent years, an interest in generalizing convolution to non-Euclidean domains such as graphs and 3D surfaces has led to the incorporation of additional group symmetries into deep learning architectures [5], [6]. In another line of work, Kakarala [7] and Kondor [8] pioneered the use of the analytical group-invariant bispectrum in signal processing and machine learning contexts. Both of these approaches require specifying the group of transformations *a priori* and explicitly building its structure into the network architecture. However, the groups that structure natural data are often either unknown or too complex to specify analytically—for example, when the structure arises from the interaction of many groups, or when the group acts on latent features in data.

Rather than building in groups by hand, another line of work has sought to *learn* underlying group structure solely from symmetries contained in data. The majority of these approaches use structured models that parameterize irreducible representations or Lie algebra generators, which act on the data through the group exponential map [9]–[14]. In these models, the objective is typically to infer the group element that acts on a template to generate the observed data, with inference accomplished through Expectation Maximization or other Bayesian approaches. A drawback of these models is that both the exponential map and inference schemes are computationally expensive. Thus, they are difficult to integrate into large-scale systems. A recent feed-forward approach [15] learns distributions over the group of 2D affine transformations in a deep learning architecture. However, the group is not learned in entirety, as it is restricted to a distribution on a pre-specified group.

Here, we present a novel approach for learning groups from data in an efficient, interpretable, and fully feed-forward model that requires no prior knowledge of the group, computation of the exponential map, or Bayesian inference. The key insight in our approach is to harness the generality of the form of the group-invariant bispectrum, which can be defined for arbitrary compact groups. Here, we focus on the class of compact commutative groups.

## 2 THE BISPECTRUM

The theory of groups and their representations provides a natural framework for constructing computational primitives for robust machine learning systems. We provide a one-page introduction to these mathematical foundations in Appendix A. Extensive treatment of these concepts can be found in the textbooks of Hall [16] and Gallier & Quaintance [17]. We now define the concepts essential to this work: invariance, equivariance, orbits, complete invariance, and irreducible representations.

Let $G$ be a group, $X$ a space on which $G$ acts, and $f$ a signal defined on the domain $X$. The *orbit* of a signal is the set generated by acting on the domain of the signal with each group element, i.e. $\{f(gx) : g \in G\}$. In the context of image transformations this is the set of all transformed versions of a canonical image—for example, if $G$ is the group $SO(2)$ of 2D rotations, then the orbit contains all rotated versions of that image.

A function $\phi : X \to Y$ is *G-equivariant* if $\phi(gx) = g'\phi(x)$ for all $g \in G$ and $x \in X$, with $g' \in G'$ homomorphic to $G$. That is, a group transformation on the input space results in a corresponding

group transformation on the output space. A function $\phi : X \to Y$ is *G-invariant* if $\phi(x) = \phi(gx)$ for all $g \in G$ and $x \in X$. This means that a group action on the input space has no effect on the output. We say $\phi$ is a *complete invariant* when $\phi(x_1) = \phi(x_2)$ if and only if $x_2 = gx_1$ for some $g \in G$ and $x_1, x_2 \in X$. A complete invariant collapses each orbit to a single point while also guaranteeing that different orbits map to different points. In the context of rotation-invariant vision, for example, this would mean that the only information lost through the map $\phi$ is that of the specific "pose" of the object.

A *representation* of a group G is a map $\rho : G \to GL(V)$ that assigns elements of $G$ to elements of the group of linear transformations (e.g. matrices) over a vector space $V$ such that $\rho(g_1 g_2) = \rho(g_1)\rho(g_2)$. That is, $\rho$ is a group homomorphism. In our case, $V$ is $\mathbb{C}^n$. A representation is *reducible* if there exists a change of basis that decomposes the representation into a direct sum of other representations. An *irreducible representation* is one that cannot be further reduced in this manner, and the set $Irr(G)$ are often simply called the *irreps* of $G$. The irreducible representations of all finite commutative groups are one-dimensional (therefore equivalent to the characters of the irreps), and in bijection with the group itself.

## 2.1 FOURIER ANALYSIS ON GROUPS

A powerful approach for constructing invariants leverages the group theoretic generalization of the classical Fourier Transform [18]. Let $f(g) : G \to \mathbb{R}$ be a signal on a group $G$, and let $Irr(G)$ denote the irreducible representations of the group. We assume that $\rho \in Irr(G)$ are also *unitary* representations (i.e., $\rho^{-1} = \rho^\dagger$, with $\rho^\dagger$ the conjugate transpose). The *Generalized Fourier Transform* (GFT) is the linear map $f \mapsto \hat{f}$, in which Fourier frequencies are indexed by $\rho \in Irr(G)$:

$$\hat{f}_\rho = \sum_{g \in G} \rho(g) f(g). \tag{1}$$

Readers familiar with the classical Fourier transform should identify $\rho(g)$ with the exponential term in the usual sum or integral. Here, they arise naturally as the representations of $\mathbb{Z}/n\mathbb{Z}$ or $S^1$. Intuitively, these $\rho$ are the "fundamental frequencies" of the group. In particular, for $G = \mathbb{Z}/n\mathbb{Z}$—the group of *integers modulo n*—they are the classical $n$ discrete Fourier frequencies: $\rho_k(g) = e^{-\mathbf{i}2\pi kg/n}$, for $k = 0, \ldots, n-1$ and $g \in \mathbb{Z}/n\mathbb{Z}$. For compact continuous groups $G$, the sum above is replaced by an integral $\int \rho(u) f(u) du$ over a Haar measure $du$ for the group.

Importantly, the Fourier transform is *equivariant* to the action of translation on the domain of the input—a fundamental, if not defining, property. That is, if $f^t = f(g - t)$ is the translation of $f$ by $t \in G$, its Fourier transform becomes:

$$(\hat{f^t})_\rho = \rho(t)\hat{f}. \tag{2}$$

For the classical Fourier transform, this is the familiar *Fourier shift property*: translation of the input results in phase rotations of the complex Fourier coefficients. This property is exploited in one well-known Fourier invariant—the *power spectrum*:

$$q_\rho = \hat{f}_\rho^\dagger \hat{f}_\rho. \tag{3}$$

The power spectrum preserves only the magnitude of the Fourier coefficients, eliminating the phase information entirely by multiplying each coefficient by its conjugate:

$$q_k^t = \left(e^{-\mathbf{i}2\pi kt/n} \hat{f}_k\right)^\dagger \left(e^{-\mathbf{i}2\pi kt/n} \hat{f}_k\right) = e^{\mathbf{i}2\pi kt/n} e^{-\mathbf{i}2\pi kt/n} \hat{f}_k^\dagger \hat{f}_k = q_k.$$

Although this approach eliminates variations due to translations of the input, it also eliminates much of the signal structure, which is largely contained in relative phase relationships [19]. Thus, the power spectrum is *incomplete*. As a result, it is easy to create "adversarial examples" demonstrating the incompleteness of this invariant: any two images with identical Fourier magnitudes but non-identical phase have the same power spectra (Appendices C and D).

## 2.2 THE BISPECTRUM

The *bispectrum* is a lesser known Fourier invariant parameterized by pairs of frequencies in the Fourier domain. The translation-invariant *bispectrum* for 1D signals $f \in \mathbb{R}^n$ is the complex matrix $B \in \mathbb{C}^{n \times n}$:

$$B_{i,j} = \hat{f}_i \hat{f}_j \hat{f}_{i+j}^\dagger, \tag{4}$$

where $\hat{f} = (\hat{f}_0, \ldots, \hat{f}_{n-1})$ is the 1D Fourier transform of $f$ and the sum $i + j$ is taken modulo $n$. Intuitively, bispectral coefficients $B_{ij}$ indicate the strength of couplings between different phase

components in the data, preserving the phase structure that the power spectrum loses. Again, the equivariance property of the GFT is key to $B$'s invariance. Like the power spectrum, it cancels out phase shifts due to translation. However it does so while preserving the relative phase structure of the signal. This remarkable property may be seen as follows:

$$
\begin{aligned}
B_{k_1,k_2}^t &= e^{-i2\pi k_1 t/n}\hat{f}_{k_1}e^{-\mathbf{i}2\pi k_2 t/n}\hat{f}_{k_2}e^{\mathbf{i}2\pi(k_1+k_2)t/n}\hat{f}_{k_1+k_2}^{\dagger} \\
&= e^{-\mathbf{i}2\pi k_1 t/n}e^{-\mathbf{i}2\pi k_2 t/n}e^{\mathbf{i}2\pi k_1 t/n}e^{\mathbf{i}2\pi k_2 t/n}\hat{f}_{k_1}\hat{f}_{k_2}\hat{f}_{k_1+k_2}^{\dagger} = B_{k_1,k_2}.
\end{aligned}
$$

Unlike the power spectrum, the bispectrum (for compact groups) is complete and can be used to reconstruct the original signal up to translation [20]. More generally, the bispectrum is defined for arbitrary compact groups:

$$
B_{\rho_i,\rho_j} = \hat{f}_{\rho_i}\hat{f}_{\rho_j}\hat{f}_{\rho_i\rho_j}^{\dagger}, \tag{5}
$$

with the addition modulo $n$ replaced by the group product. For non-commutative groups, the bispectrum has a similar structure, but is modified to accommodate the fact that the irreps of non-commutative groups map to matrices rather than scalars (Appendix B).

Historically, the bispectrum emerged from the study of the higher-order statistics of non-Gaussian random processes, and was used as a tool to measure the non-Gaussianity of a signal [21]–[23]. The work of Yellott and Iverson [20] established that every integrable function with compact support is completely identified—up to translation—by its three-point autocorrelation and thus its bispectrum (Appendix F). Later work of Kakarala tied the bispectrum to the more general formulation of Fourier analysis and thus defined it for all compact groups [4], [7]. Kakarala went on to prove that it is complete [24] in both commutative and non-commutative formulations (Appendix D).

## 3 BISPECTRAL NEURAL NETWORKS

We draw inspiration from the group-invariant bispectrum to define Bispectral Neural Networks, a neural network architecture that parameterizes the *ansatz* of the bispectrum. The key idea behind this approach is to constrain the optimization problem by building in the *functional form* of a solution known to exist. Importantly, we build in only the generic functional form and allow the network to learn the appropriate parameters (and thus the group over which it is defined) from data. We draw upon two key properties of the bispectrum, which allow it to be naturally parameterized and learned:

1. The bispectrum is computed on a group-equivariant Fourier transform. The Fourier transform is a linear map and can be written as $z = Wx$, with $W$ a matrix of irreducible representations—the Discrete Fourier Transform (DFT) matrix, in the classical case. If the group is unknown, we can treat $W$ as a matrix of parameters to be learned through optimization. The bispectrum computation is identical regardless of the structure of the irreps; thus we can use a single network architecture to learn finite approximations of arbitrary compact commutative groups.

2. The bispectrum separates orbits. Here, we use orbit separation at the output of the bispectrum computation as the criterion to optimize the parameters defining the group-equivariant Fourier transform. This requires knowing only which datapoints are in the same orbit (but not the group that generates the orbit), and is similar in spirit to contrastive learning [25].

### 3.1 MODEL ARCHITECTURE

Concretely, given an input vector $x \in \mathbb{R}^n$, the Bispectral Neural Network module is completely specified by one complex matrix $W \in \mathbb{C}^{n \times n}$, where each row $W_i \in \mathbb{C}^n$ plays the role of an irreducible representation. This matrix defines the linear transformation:

$$
z = Wx. \tag{6}
$$

If $W$ defines a Fourier transform on $G$, then each element of $z$ is a coefficient on a different irreducible representation of $G$. The analytical bispectrum can then be computed from triples of coefficients as:

$$
B_{i,j} = z_i z_j z_{ij}^{\dagger} \tag{7}
$$

However, if the group is unknown, then we do not know the product structure of the group and thus do not know *a priori* which element in the vector $z$ corresponds to the $(ij)^{\text{th}}$ irrep. Conveniently, for commutative groups, all irreducible representations are in bijection with the group elements. That is,

each irreducible representation can be uniquely identified with a group element. Consequently, the $(ij)^{\text{th}}$ representation can obtained from the element-wise multiplication of the $i^{\text{th}}$ and $j^{\text{th}}$ irreducible representations. Thus, the (conjugated) $(ij)^{\text{th}}$ coefficient can be obtained as

$$z_{ij}^{\dagger} = (W_i \odot W_j)^{\dagger} x, \tag{8}$$

with $\odot$ indicating the Hadamard (element-wise) product and $^{\dagger}$ the complex conjugate. Plugging this into Equation 7 and expanding the terms, we obtain the Bispectral Network:

$$\beta_{i,j}(x) = W_i x \cdot W_j x \cdot (W_i \odot W_j)^{\dagger} x \tag{9}$$

Figure 1 illustrates the computation. Each linear term $W_i x$ is an inner product yielding a scalar value, and the three terms are combined with scalar multiplication in $\mathbb{C}$. Note that this equation shows the computation of a single scalar output $\beta_{i,j}$. In practice, this is computed for all non-redundant pairs.

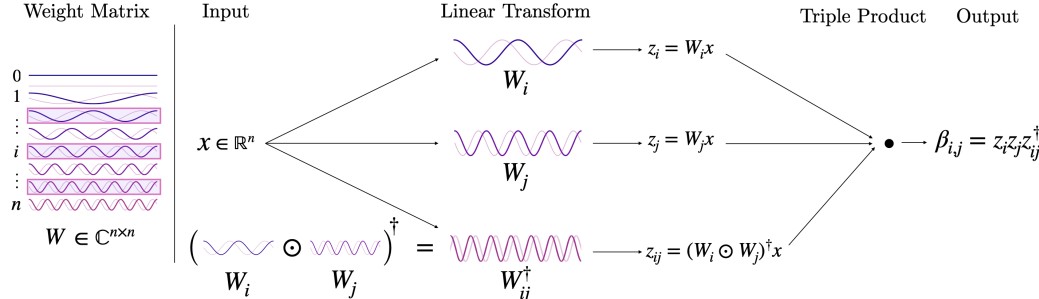

**Figure 1: Bispectral Neural Networks**. A single linear neural network layer parameterized by $W$ generates the output $z = Wx$. Pairs of coefficients $z_i$ and $z_j$ and the corresponding weights $W_i$ and $W_j$ are then used to compute the output of the network $\beta_{i,j} = z_i z_j z_{ij}^{\dagger}$, with $z_{ij}$ computed as $(W_i \odot W_j)x$. Here, the weights are depicted as the canonical Fourier basis on $\mathbb{S}^1$ for illustrative purposes. In practice, the weights are randomly initialized, but are expected to converge to the irreps of the group that structures the dataset.

While the bispectrum is a third-order polynomial, it results in only $n^2$ rather than $n^3$ products, due to the constraint on the third term. Moreover, the bispectrum has many symmetries and is thus highly redundant. The most obvious symmetry is reflection over the diagonal—i.e. $\beta_{i,j} = \beta_{j,i}$. However, there are many more (see Appendix G). Here, we compute only the upper triangular of the matrix.

As the Bispectral Network computes third-order products of the input, the output values can be both very large and very small, which can result in numerical instability during optimization. To combat this, we normalize the output of the network to the unit sphere in $\mathbb{C}^n$:

$$\bar{\beta}(x) = \frac{\beta(x)}{||\beta(x)||_2} \tag{10}$$

We demonstrate in Appendix E that the analytical bispectrum can be normalized to the unit sphere while preserving completeness up to a scalar factor (Appendix E, Theorem E.3). Thus, we similarly expect that this normalization will not impact the robustness of the network.

## 3.2 Orbit Separation Loss

Given a dataset $X = \{x_1, ..., x_m\}$ with latent group structure and labels $Y = \{y_1, ..., y_m\}$ indicating which datapoints are equivalent up to transformation (but without information about the transformation itself, i.e. *how* they are related), we wish to learn an invariant map that collapses elements of the same group orbit and separates distinct orbits. Critically, we want this map to be both *invariant* to group actions and *selective* to pattern structure—that is, to be *complete* [24]. A network that successfully learns such a map will have implicitly *learned the group* that structures transformations in the data.

To that end, we introduce the *Orbit Separation Loss*. This loss encourages the network to map elements of the same orbit to the same point in the representation space, while ensuring degenerate solutions (i.e. mapping all inputs to the same point) are avoided:

$$L(x_a) = \sum_{j|y_b = y_a} ||\bar{\beta}(x_a) - \bar{\beta}(x_b)||_2^2 + \gamma ||x_a - W^{\dagger} W x_a||_2^2. \tag{11}$$

The first term measures the Euclidean distance in the output space between points in the same orbit. Thus, minimizing the loss encourages them to be closer. In the second term, the input $x_a$ is reconstructed by inverting the linear transform as $W^{\dagger} W x_a$. The second term is thus a measure of

whether information is preserved in the map. During training, each row vector in the weight matrix $W$ is normalized to unit length after every gradient update, which pushes $W$ towards the manifold of unitary matrices. The coefficient $\gamma$ weights the contribution of the reconstruction error to the loss.

## 4 EXPERIMENTS AND ANALYSES

We now test the capacity of Bispectral Networks to learn a group by learning to separate orbit classes. We conduct four experiments to analyze the properties of the architecture: learning Fourier transforms on groups (Section 4.1), group-invariant classification (Section 4.2), model completeness (Section 4.3), and extracting the Cayley table (Section 4.4).

### 4.1 LEARNING FOURIER TRANSFORMS ON GROUPS

We first examine whether Bispectral Networks trained to separate orbit classes will learn the Fourier transform and corresponding bispectrum of the group that structures the data. Here, we test the network on two groups, which act on the image grid to generate image orbits. The first is one whose bispectrum is known and well-understood: the group $\mathbb{S}^1 \times \mathbb{S}^1$ of 2D cyclic translations, i.e. the canonical 2D Fourier transform. The second is the group $SO(2)$. To our knowledge, a bispectrum on $SO(2)$ acting on the grid (or disk) has not been defined in the mathematics literature, as the domain is not a homogeneous space for the group—an assumption fundamental to much of bispectrum theory. However, a Fourier transform for $SO(2)$ acting on the disk has been defined [26] using the orthogonal basis of disk harmonics. We thus expect our weight matrix $W$ to converge to the canonical Fourier basis for the group $\mathbb{S}^1 \times \mathbb{S}^1$, and hypothesize the emergence of disk harmonics for $SO(2)$.

Models are trained on datasets consisting of 100 randomly sampled (16, 16) natural image patches from the van Hateren dataset [27] that have been transformed by the two groups to generate orbit classes. Further details about the dataset can be found in Appendix H. Networks are trained to convergence on the Orbit Separation Loss, Equation 11. All hyperparameters and other details about the training procedure can be found in Appendix I.1.

Remarkably, we find that the relatively loose constraints of the orbit separation loss and the product structure of the architecture is sufficient to encourage the network to learn a group-equivariant Fourier transform and its corresponding bispectrum from these small datasets. This is observed in the structure of the network weights, which converge to the irreducible representations of the group in the case of $\mathbb{S}^1 \times \mathbb{S}^1$, and filters resembling the disk harmonics for $SO(2)$) (Figure 2). Importantly, we observe that the learned linear map (Equation 6) is equivariant to the action of the group on the input, and the output of the network (Equation 10) is invariant to the group action on arbitrary input signals. We visualize this in Figure 3 for transformed exemplars from the MNIST dataset, which the model was not trained on, demonstrating the capacity of the model to generalize these properties to out-of-distribution data.

$$\mathbb{S}^1 \times \mathbb{S}^1 \qquad SO(2)$$

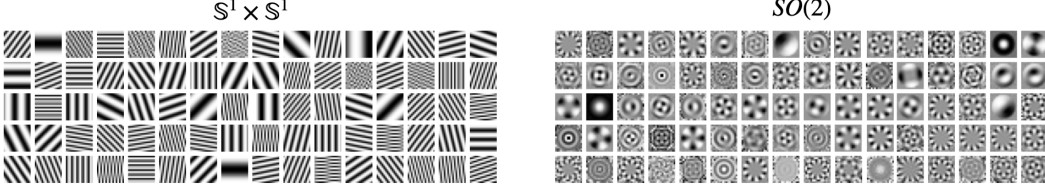

**Figure 2: Filters Learned in Bispectral Networks**. Real components of weights learned in the Bispectral Network trained on the transformed van Hateren Natural Images datasets, for $\mathbb{S}^1 \times \mathbb{S}^1$ 2D Translation (left) and $SO(2)$ 2D Rotation dataset (right). The full set of learned weights can be found in Appendix J.

### 4.2 GROUP-INVARIANT CLASSIFICATION

We next examine the utility of the rotation model learned in Section 4.1 for facilitating performance on a downstream image classification task: digit classification for the Rotated MNIST dataset. For this experiment, we append a simple multi-layer fully-connected ReLU network (MLP; the "classification network") to the output of the Bispectral Network and train it on the classification objective. While the Bispectral Network learns to eliminate factors of variation due to group actions, the classification network learns to eliminate other variations within digit classes irrelevant to classification, i.e. those due to handwriting style. We train the classification model using the scheme described in Appendix I. We compare the performance of our model to four leading approaches to group-invariant classification, in addition to a Linear Support Vector Machine (SVM) model that we train as a baseline (Table 1).

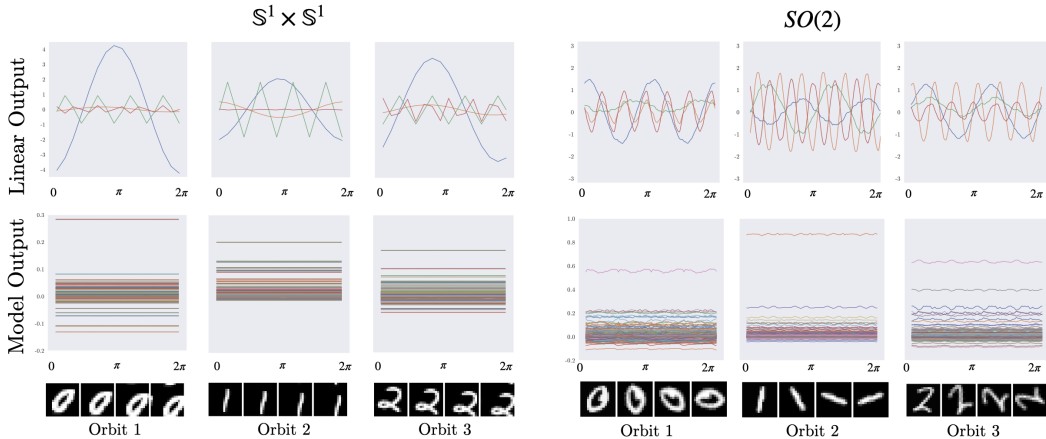

**Figure 3: Model Outputs on Transformed Data.** Real components of the outputs of the linear transform ($Wx$, top row) and full network output (Eq. 10, bottom row). Data inputs are cyclically translated/rotated in one direction (x-axis) and layer/network outputs are computed for each transformation. Each colored line represents the output of a single neuron, with the activation value on the y-axis. In the top row, only four neuron outputs are visualized, for clarity. The equivariance of the first layer is observed in the sinuisoidal responses, which reveal equivariant phase shifts on each neuron's output as the input transforms. The invariance of the network output is observed in the constant response across all translated or rotated images.

Our model obtains a competitive 98.1% classification accuracy on this task—within 0.2 - 1.2% of the state-of-the-art methods compared to here. It also possesses several notable differences and advantages to the models presented in this table. First, our model *learns the group* from data, in contrast to Harmonic Networks and E2CNNs—which build in $SO(2)$-equivariance—and Augerino, which learns a distribution over the pre-specified group of 2D affine transformations. Although our method is marginally outperformed by these models in terms of classification accuracy, ours outperforms [13], the only other model that fully learns a group. Second, our model is fully feed-forward, and it does not rely on the computationally intensive EM inference scheme used in [13], or the exponential map as used in [13] and [15]. Finally, in the next section, we demonstrate that our model is approximately *complete*. By contrast, we demonstrate, representations within the leading models E2CNN and Augerino exhibit excessive invariance.

| Model | Learns Group | Fully Feedforward | Approximately Complete | RotMNIST Accuracy |
|---|---|---|---|---|
| Linear SVM (Baseline) | N | NA | - | 54.7% |
| Cohen & Welling (2014) [13] | **Y** | N | - | $\sim 97\%^{\dagger}$ |
| Harmonic Networks (2017) [28] | N | **Y** | - | 98.3% |
| Augerino (2020) [15] | **Y**$^*$ | **Y** | N | 98.9% |
| E2CNN (2019) [29] | N | **Y** | N | **99.3**% |
| **Bispectral Network + MLP** | **Y** | **Y** | **Y** | 98.1% |

**Table 1: Model Comparison on RotMNIST**. Bispectral Networks achieve 98.1% classification accuracy on RotMNIST. The model is marginally outperformed by three models that require the partial or complete specification of the group structure in their design. Our model, by contrast *learns* the group entirely from data. Further, our model is approximately complete, while the leading models E2CNN and Augerino exhibit excessive invariance (Figure 4).

$^{\dagger}$This number was estimated from the bar plot in Figure 4 of [13] and approved by the first author of the paper in personal communication.
$^*$ The model in [15] learns a *distribution* over a pre-specified group. It is different in kind to [13] and our model, which learn the group itself.

## 4.3 Model Completeness

A key property of the analytical bispectrum is its *completeness*: only signals that are exactly equal up to the group action will map to the same point in bispectrum space. Here, we analyze the completeness of the trained models, using *Invariance-based Adversarial Attacks* (IAAs) [2] as a proxy. IAAs aim to identify *model metamers*—inputs that yield identical outputs in the model's representation space, but are perceptually dissimilar in the input space. For an approximately complete model, we expect

all inputs that map to the same point in model representation space to belong to the same orbit. That is, the only perturbations in pixel space that will give rise to identical model representations are those due to the group action.

We test the completeness of the trained models as follows. A batch of targets $\{x_1, ..., x_N\}, x \in \mathbb{R}^m$ are randomly drawn from the RotMNIST dataset. A batch of inputs $\{\bar{x}_1, ..., \bar{x}_N\}$, $\bar{x} \in \mathbb{R}^m$ are randomly initialized, with pixels drawn from a standard normal distribution. Both the targets and random inputs are passed through the network to yield $\bar{\beta}(x_i)$ (target) and $\bar{\beta}(\bar{x}_i)$ (adversary). The following objective is minimized with respect to the $\bar{x}_i$:

$$L = \sum_{i=0}^{N} ||\bar{\beta}(x_i) - \bar{\beta}(\bar{x}_i)||_2^2 \tag{12}$$

An optimized input $\bar{x}_i$ is considered a model metamer if it yields a model representation within $\epsilon$ of $\bar{\beta}(x_i)$, is classified equivalently as $x_i$, but is not in the orbit of $x_i$. Here, we aim to minimize distance between targets and adversaries in the *model representation space*—i.e. the output of the Bispectral Network, prior to the classification model. For the comparison models E2CNN and Augerino, we use the output of the layer prior to the softmax.

Figure 4 shows the targets and the optimized adversarial inputs for the Bispectral Network trained on rotation in the van Hateren dataset, and the E2CNN and Augerino models trained on Rotated MNIST. For the Bispectral Network, we observe that all random initializations consistently converge to an element within the orbit of the target, indicating that the invariant map learned by the model is close to complete. By contrast, many unrelated or noisy metameric images can be found for E2CNN and Augerino. These metamers yield identical model representations (up to $\epsilon$) and identical classifications as the targets, but are not contained in their orbits of the targets. Interestingly, the metamers optimized for Augerino possess structure that at times resembles features in the target image. We roughly estimate the percentage of these "perceptually similar" inputs as $35\%$. The model nonetheless exhibits excessive invariance and is not complete.

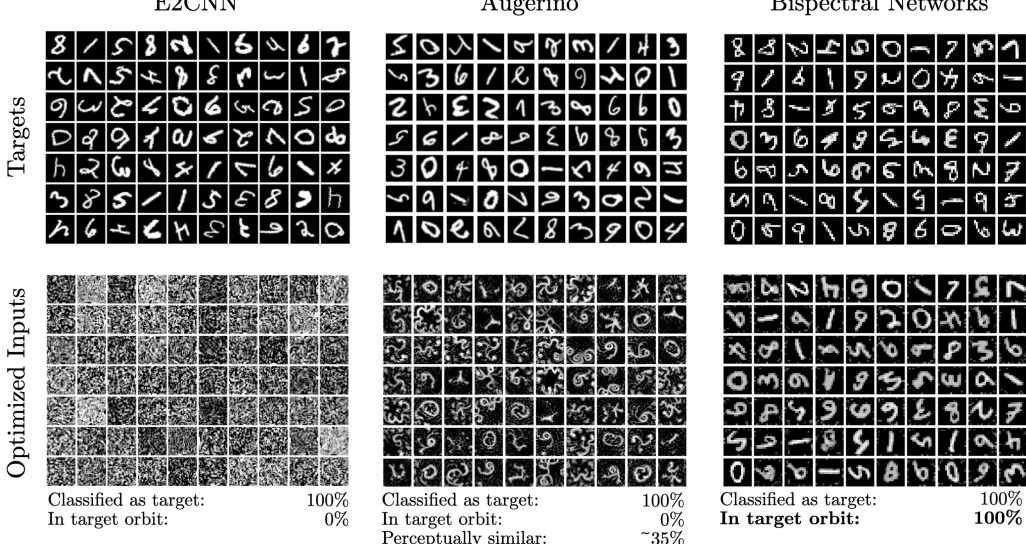

**Figure 4: Completeness in Bispectral Networks.** Targets and optimized inputs for (left) E2CNN, (middle) Augerino, and (right) Bispectral Networks. Images are randomly initialized and optimized to yield representations identical to a target image. If a model possesses excessive invariance, there exist images that will yield equivalent representations while being non-equivalent up to the group action. For Bispectral Networks, all optimized inputs are within the orbits of the targets, indicating approximate completenes. By contrast, many model metamers are found for E2CNN and Augerino.

## 4.4 Learning the Group: Extracting the Cayley Table

Finally, we demonstrate that the converged network weights can be used to *construct the group itself*—i.e. the product structure of the group, as captured by the group's Cayley table. For finite groups, a Cayley table provides a complete specification of group structure. Here, we demonstrate a

novel method for constructing a Cayley table from learned irreps, described in in Appendix K, which exploits the bijection between irreps and group elements in commutative groups.

We test this approach on all unique finite commutative groups of order eight: $\mathbb{Z}/8\mathbb{Z}$, $\mathbb{Z}/4\mathbb{Z} \times \mathbb{Z}/2\mathbb{Z}$, and $\mathbb{Z}/2\mathbb{Z} \times \mathbb{Z}/2\mathbb{Z} \times \mathbb{Z}/2\mathbb{Z}$. For each group, we generate a synthetic dataset consisting of 100 random functions over the group, $f : G \to \mathbb{R}$, with values in $\mathbb{R}$ drawn from a standard normal distribution, and generate the orbit of each exemplar under the group. Networks are trained to convergence on each dataset using the Orbit Separation Loss, Equation 11. Details about the training procedure can be found in Appendix I.4. After a model has converged, we use its learned weights to compute a Cayley table using Algorithm 1 in Appendix K. We check the isomorphism of the learned Cayley table with the ground truth using Algorithm 3 in Appendix K.

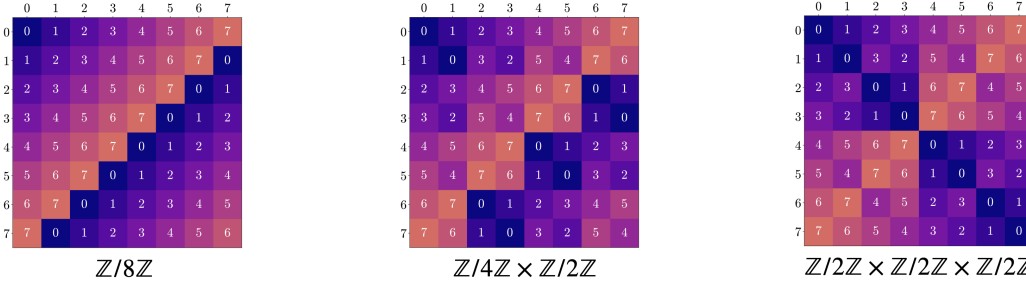

$$\mathbb{Z}/8\mathbb{Z} \qquad\qquad \mathbb{Z}/4\mathbb{Z} \times \mathbb{Z}/2\mathbb{Z} \qquad\qquad \mathbb{Z}/2\mathbb{Z} \times \mathbb{Z}/2\mathbb{Z} \times \mathbb{Z}/2\mathbb{Z}$$

**Figure 5: Learned Cayley Tables for All Unique Groups of Order Eight**. Rows and columns are labeled with integers that index group elements. Each cell of the table contains the index of the group element obtained by composing the group elements indexed in the row and column. The Cayley table thus contains all information about the group structure. Bispectral Networks perfectly recover the Cayley tables for all groups tested here.

Strikingly, we are able to recover the exact structure of the groups that generated the data orbits from the converged models' weights, in the form of the Cayley tables displayed in Figure 5. For each dataset, the recovered Cayley tables are identical to the known ground-truth Cayley tables. To our knowledge, this is the first demonstration that an explicit representation of the product structure of an unknown group can be learned from data. Our results on all finite commutative groups of order eight suggest that this will generalize more broadly to all finite commutative groups.

## 5 DISCUSSION

In this work, we introduced a novel method for learning groups from data in a neural network architecture that simultaneously learns a group-equivariant Fourier transform and its corresponding group-invariant bispectrum. The result is an interpretable model that learns the irreducible representations of the group, which allow us to recover an explicit model of the product structure of the group—i.e. the group itself, in the form of its Cayley table. The resulting trained models are both powerful and robust—capable of generalizing to arbitrary unseen datasets that are structured by the same transformation group, while exhibiting approximate *completeness*. To our knowledge, these results provide the first demonstration that a bispectrum or Cayley table can be learned simply from the symmetries in data alone. We view the work presented here as a starting point for learning the more complex groups needed to recognize 3D objects in natural imagery. In ongoing work, we are extending the model to the non-commutative form of the bispectrum (Appendix B), and developing a localized version amenable to stacking hierarchically in deep networks.

The precise convergence of the Bispectral Network parameters to canonical forms from representation theory—under the relatively weak constraints of the third-order model ansatz and the orbit separation loss—suggests the uniqueness of this solution within the model class. Indeed, the Fourier solution is likely guaranteed by the uniqueness of the bispectrum as a third-order polynomial invariant [30]. Future work should explore the bounds of these guarantees in the context of statistical learning. We venture to suggest further that the uniqueness and completeness of the bispectrum make it a compelling candidate computational primitive for invariant perception in artificial and biological systems alike. Indeed, results from across areas of visual cortex suggest that group representations may play an important role in the formation of visual representations in the brain [31]. Models like the one presented here may provide concrete hypotheses to test in future mathematically motivated neuroscience research.

## REPRODUCIBILITY STATEMENT

The code to implement all models and experiments in this paper can be found at `github.com/sophiaas/bispectral-networks`. We provide files containing the trained models, configs containing the hyperparameters used in these experiments, notebooks to generate the visualizations in the paper, and download links for the datasets used in these experiments.

## ACKNOWLEDGMENTS

The authors thank Nina Miolane, Khanh Dao Duc, Giovanni Luca Marchetti, David Klindt, Mathilde Papillon, Adele Myers, Taco Cohen, and Ramakrishna Kakarala for helpful suggestions, feedback, and proofreading.

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

## APPENDIX A   MATHEMATICAL BACKGROUND

**Groups.** A *group* $(G, \cdot)$ is a set $G$ with a binary operation $\cdot$, which we can generically call the *product*. The notation $a \cdot b$ denotes the product of two elements in the set; however, it is standard to omit the operator and write simply $ab$. Concretely, a group $G$ may define a class of transformations, such as two-dimensional translations or rotations in the plane. The elements of the group $g \in G$ define *particular* transformations, such as *rotation by* $30°$ or *rotation by* $90°$. The binary operation $\cdot$ provides a means for combining two particular transformations—for example, first rotating by $30°$ and then rotating by $90°$. For a set of transformations $G$ to be a group under the operation $\cdot$, the four following axioms must hold:

1. *Closure*: The product of any two elements of the group is also an element of the group, i.e. for all $a, b \in G$, $ab \in G$.

2. *Associativity*: The grouping of elements under the operation does not change the outcome, so long as the order of elements is preserved, i.e. $(ab)c = a(bc)$.

3. *Identity*: There exists a "do-nothing" *identity* element $e$ that such that the product of $e$ with any other element $g$ returns $g$, i.e. $ge = eg = g$ for all $g \in G$.

4. *Inverse*: For every element $g$, there exists an *inverse* element $g^{-1}$ such that the product of $g$ and $g^{-1}$ returns the identity, i.e. $gg^{-1} = g^{-1}g = e$.

**Homomorphisms.** Two groups $(G, \cdot)$ and $(H, *)$ are *homomorphic* if there exists a correspondence between elements of the groups that respect the group operation. Concretely, a *homomorphism* is a map $\rho : G \to H$ such that $\rho(u \cdot v) = \rho(u) * \rho(v)$. An *isomorphism* is a bijective homomorphism.

**Product Groups.** A *product group* as a set is the cartesian product $G \times H$ of two given groups $G$ and $H$. The new group product is given by $(g_1, h_1) \cdot (g_2, h_2) = (g_1 g_2, h_1 h_2)$.

**Commutativity.** A group $(G, +)$ is *commutative* or *abelian* if the order of operations does not matter, i.e. $ab = ba$. If this does not hold for all elements of the group, then the group is called *non-commutative*. The classification of finite commutative groups says that each such group is a product of cyclic groups.

**Group Actions.** A *group action* is a map $T : G \times X \to X$ that maps $(g, x)$ pairs to elements of $X$. We say a group $G$ *acts* on a space $X$ if the following properties of the action $T$ hold:

1. The identity $e \in G$ maps an element of $x \in X$ to itself, i.e. $T(e, x) = x$

2. Two elements $g_1, g_2 \in G$ can be combined before or after the map to yield the same result, i.e. $T(g_1, T(g_2, x)) = T(g_1 g_2, x)$

For simplicity, we will use the shortened notation $T_g x$ to denote $T(g, x)$, often expressed by saying that a point $x$ maps to $gx (= T_g(x))$.

**Invariance**. A function $\phi : X \mapsto Y$ is *G-invariant* if $\phi(x) = \phi(gx)$ for all $g \in G$ and $x \in X$. This means that group actions on the input space have no effect on the output.

**Equivariance**. A function $\phi : X \mapsto Y$ is *G-equivariant* if $\phi(gx) = g'\phi(x)$ for all $g \in G$ and $x \in X$, with $g' \in G'$, a group homomorphic to $G$ that acts on the output space. This means that a group action on the input space results in a corresponding group action on the output space.

**Orbits.** Given a point $x \in X$, the *orbit $Gx$* of $x$ is the set $\{gx : g \in G\}$. In the context of image transformations, the orbit defines the set of all transformed versions of a canonical image—for example, if $G$ is the group of translations, then the orbit contains all translated versions of that image.

**Homogeneous Spaces.** We say that $X$ is a *homogeneous space* for a group $G$ if $G$ acts transitively on $X$—that is, if for every pair $x_1, x_2 \in X$ there exists an element of $g \in G$ such that $gx_1 = x_2$. A homogeneous space $X$ equipped with the action of $G$ is called a **G**-*space*. The concept can be clearly illustrated by considering the surface of a sphere, the space $S^2$. The group $SO(3)$ of orthogonal $3 \times 3$ matrices with determinant one defines the group of 3-dimensional rotations. The sphere $S^2$ is a homogeneous space for $SO(3)$. For every pair of points on the sphere, one can define a 3D rotation matrix that takes one to the other.

**Representations.** A *representation* of a group G is a map $\rho : G \to GL(V)$ that assigns elements of $G$ to elements of the group of linear transformations over a vector space $V$ such that $\rho(g_1 g_2) = \rho(g_1)\rho(g_2)$. That is, $\rho$ is a group homomorphism. In many cases, $V$ is $\mathbb{R}^n$ or $\mathbb{C}^n$. A representation is *reducible* if there exists a change of basis that decomposes the representation into a direct sum of other representations. An *irreducible representation* is one that cannot be further reduced in this manner, and the set of them $Irr(G)$ are often simply called the *irreps* of $G$. The irreducible representations of a finite commutative group are all one-dimensional and in bijection with the group itself. In particular, it is straightforward to compute $Irr(G)$ for such groups given their classification as products of cyclic groups $\mathbb{Z}/n\mathbb{Z}$.

## APPENDIX B    THE BISPECTRUM ON NON-COMMUTATIVE GROUPS

The theory of the bispectrum applies also in the setting of non-commutative groups and can be expressed in terms of the generalized Fourier transform and the (not necessarily one-dimensional) representations of the group. The more general, non-commutative form of the bispectrum is:

$$\beta_{\rho_i,\rho_j} = [\hat{f}\rho_i \otimes \hat{f}\rho_j]C_{\rho_i,\rho_j}\Big[ \bigoplus_{\rho \in \rho_i \otimes \rho_j} \hat{f}_\rho^\dagger \Big]C_{\rho_i,\rho_j}^\dagger, \tag{13}$$

where $\otimes$ is the tensor product, $\oplus$ is a direct sum over irreps, and the unitary matrix $C_{\rho_i,\rho_j}$ defines a Clebsch-Gordan decomposition on the tensor product of a pair of irreps $\rho_i, \rho_j$ [8].

## APPENDIX C    INCOMPLETENESS OF THE POWER SPECTRUM

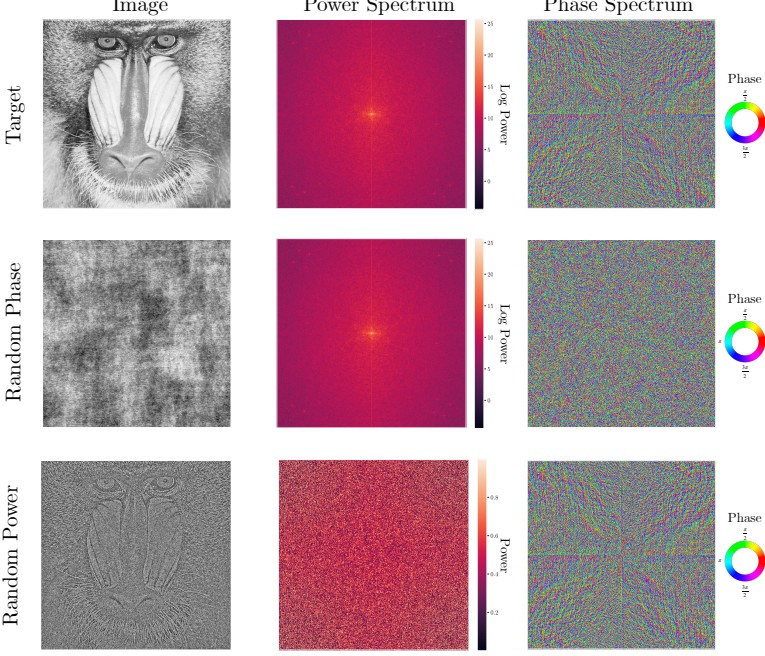

**Figure 6: The Power Spectrum is a Non-Selective Invariant.** Two images with identical power spectra (top and middle) can have very different image structure. However, images with identical phase spectra and non-identical power spectra (top and bottom) share much of the meaningful image structure—that which defines edges and contours. May require enlargement to see. Note that the sharp horizontal and vertical lines in the power and phase spectra are artifacts due to the wrap-around of the image boundaries.

## APPENDIX D    COMPLETENESS OF THE BISPECTRUM

The bispectrum is a (generically) complete invariant map with respect to a given compact group $G$ [24]. Here, *generic* is a technical term that means "outside of a set of measure zero." For instance, in the finite commutative group case, this amounts to uniqueness for all elements of the space that have all-nonzero Fourier coefficients. In practice, this property is easily derived from the exactly computable inverse of the bispectrum; see, for instance, algorithms spelled out in [32], [33]. In these computations, an inverse is defined whenever denominators—which end up being Fourier coefficients recursively computed—appearing in the calculations are nonzero.

## APPENDIX E    COMPLETENESS OF THE NORMALIZED BISPECTRUM

A useful observation is that this generalizes somewhat to a normalized form of the bispectrum, where we project features to the unit sphere in $\mathbb{C}^n$ (normalization).

**Definition E.1.** The *normalized bispectrum* of a function $f$ is defined as
$$\bar{\beta}(f) = \frac{\beta(f)}{||\beta(f)||_2}.$$

First, we show that any scalar multiple of the bispectrum preserves completeness up to positive scalar multiplication of the input signal $f$. Using this result, we show that the normalized bispectrum is generically complete up to the action of $G$ and scalar multiplication.

**Lemma E.1.** *The bispectrum of a scalar $c > 0$ times a function $f$ is:*
$$\beta(cf) = c^3 \beta(f).$$

*Proof.* By linearity of the Fourier transform, we have: $\hat{cf} = c\hat{f}$. From the definition, the commutative bispectrum of $cf$ therefore satisfies:
$$
\begin{aligned}
\beta(cf) &= c\hat{f}_i c\hat{f}_j c\hat{f}_{i+j} \\
&= c^3 \hat{f}_i \hat{f}_j \hat{f}_{i+j} = c^3 \beta(f).
\end{aligned}
\tag{14}
$$
This is easily seen to also work for the non-commutative formulation.    □

**Lemma E.2.** *If the bispectra of two (generic) functions $f$ and $g$ differ by a constant factor $c > 0$, then the orbits of the two functions differ by a constant factor, i.e.*
$$\beta(f) = c\beta(g) \implies Gf = c^{\frac{1}{3}} Gg.$$

*Proof.* Assume $\beta(f) = c\beta(g)$ and let $h = c^{\frac{1}{3}} g$. By Lemma E.1, we have: $\beta(h) = c\beta(g)$, and by assumption, $\beta(f) = c\beta(g)$ so that $\beta(f) = \beta(h)$. By (generic) completeness of the bispectrum, the orbits of $f$ and $h$ are equivalent; that is, $Gf = Gh$. Because $h$ is a scalar multiple of $g$, the orbits of $h$ and $g$ are related by a constant factor: $Gh = c^{\frac{1}{3}} Gg$. Finally, because the orbit of $f$ equals that of $h$, we see that the orbit of $f$ is related to that of $g$ by the same factor: $Gf = c^{\frac{1}{3}} Gg$.    □

**Theorem E.3.** *Two (generic) functions $f$ and $g$ have equivalent normalized bispectra if and only if their orbits under $G$ are related by a scalar factor $c \in \mathbb{R}$.*
$$\hat{\beta}(f) = \hat{\beta}(g) \iff Gf = cGg.$$

*Proof.* ($\Longleftarrow$) By assumption, $Gf = cGg$, which, together with Lemma E.1, implies $\beta(f) = c^3 \beta(g)$.

Writing out the definition of the normalized bispectrum, we see

$$
\begin{aligned}
\bar{\beta}(f) &= \frac{\beta(f)}{||\beta(f)||} \\
&= \frac{c^3 \beta(g)}{||c^3 \beta(g)||} \\
&= \frac{c^3 \beta(g)}{c^3 ||\beta(g)||} \\
&= \frac{\beta(g)}{||\beta(g)||} = \bar{\beta}(g).
\end{aligned}
\tag{15}
$$

($\Rightarrow$) Given $\bar{\beta}(f) = \bar{\beta}(g)$, we expand and see the following relation:

$$
\begin{aligned}
\bar{\beta}(f) &= \bar{\beta}(g) \\
\frac{\beta(f)}{||\beta(f)||} &= \frac{\beta(g)}{||\beta(g)||} \\
\beta(f) &= \frac{||\beta(f)||}{||\beta(g)||} \beta(g).
\end{aligned}
\tag{16}
$$

Letting $c = \frac{||\beta(f)||}{||\beta(g)||}$, we have:

$$
\beta(f) = c\beta(g),
$$

which by Lemma E.2 implies that $Gf = cGg$. $\qquad\square$

## APPENDIX F  POLYSPECTRA AND AUTOCORRELATIONS

Both the power spectrum and bispectrum are intimately related to the second- and third-order statistics of a signal, as the Fourier transforms of the two- and three-point autocorrelation functions, respectively. The two-point *autocorrelation* $A_2$ of a complex-valued function $f$ on the real line is the integral of that function multiplied by a shifted copy of it:

$$
A_{2,f}(s) = \int_{-\infty}^{\infty} f^{\dagger}(x) f(x+s) dx,
\tag{17}
$$

where $s$ denotes the increment of the shift. Taking its Fourier transform, it becomes:

$$
\hat{A}_{2,f}(s) = \hat{f}_k \hat{f}_k^{\dagger},
\tag{18}
$$

which is the power spectrum. The three-point autocorrelation, also called the *triple correlation*, is the integral of a function multiplied by two independently shifted copies of it:

$$
A_{3,f}(s_1, s_2) = \int_{-\infty}^{\infty} f^{\dagger}(x) f(x+s_1) f(x+s_2) dx.
\tag{19}
$$

Taking its Fourier transform it becomes:

$$
\hat{A}_{2,f}(s_1, s_2) = \hat{f}_{k_1} \hat{f}_{k_2} \hat{f}_{k_1+k_2}^{\dagger},
\tag{20}
$$

which is the bispectrum.

## APPENDIX G  SYMMETRIES IN THE BISPECTRUM

The bispectrum has $n^2$ terms (with $n = $ number of frequencies). However, the object is highly redundant and is reducible to $\frac{n^2}{12}$ terms without loss of information for a real-valued signal [23]. This is due to the following symmetries, depicted in Figure 7. For $\omega_1 = 2\pi k_1$:

$$\begin{aligned}
\beta_{\omega_1,\omega_2} &= \beta_{\omega_2,\omega_1} \\
&= \beta_{-\omega_2,-\omega_1} = \beta_{-\omega_1-\omega_2,\omega_2} \\
&= \beta_{\omega_1,-\omega_1-\omega_2} = \beta_{-\omega_1-\omega_2,\omega_1} \\
&= \beta_{\omega_2,-\omega_1-\omega_2}.
\end{aligned}$$

Thus, the subset of bispectral coefficients for which $\omega_2 >= 0$, $\omega_1 >= \omega_2$, $\omega_1 + \omega_2 <= \pi$ is sufficient for a complete description of the bispectrum.

Symmetries of the Triple Correlation and Bispectrum

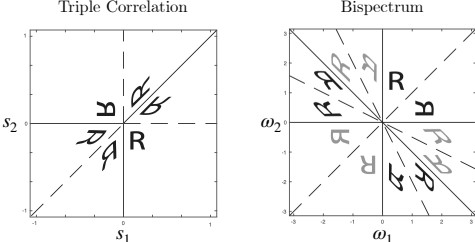

**Figure 7: Symmetries of the Triple Correlation and Bispectrum**. For a real-valued signal, the six-fold symmetry of the triple product becomes a twelve-fold symmetry in which six paris of regions are conjugate symmetric (with the complex conjugate indicated in grey). Adapted from [23].

It is also known that, in fact, the bispectrum can be further reduced. For example, for the bispectrum on 1D translation, only $n + 1$ coefficients are needed to reconstruct the input up to translation, and thus define a complete description of the signal structure in the quotient space [33]. The reconstruction algorithm proposed in [33] requires that the signal has no Fourier coefficients equal to zero. More robust reconstruction algorithms, such as least squares approaches, use more elements of the bispectrum matrix [34] for stability purposes.

## APPENDIX H  DATASETS

### H.1  VAN HATEREN NATURAL IMAGES

We use natural image patches from the Van Hateren database [27] as the data on which the groups act, and generate two datasets, each using a different group action: Cyclic 2D Translation ($S^1 \times S^1$) and 2D Rotation $SO(2)$. For each group, we randomly select 100 image patches as the "exemplars" on which the group acts. Each image patch is augmented by each element of the group to generate the orbit of the image under the transformation. The group $S^1 \times S^1$ is discretized into integer translations, and all 256 possible discrete translations are used to generate the data. The group $SO(2)$ is discretized into 360 integer degree rotations and 30% of these rotations are randomly drawn for each image to generate its orbit. Following orbit generation, image patches are normalized to have zero mean and unit standard deviation. For the 2D rotation dataset, patterns are cropped to a circle with diameter equal to the side length of the image, with values outside of the circle set to zero, to maintain perfect rotational symmetry. Appendix H shows examples from the dataset. A random 20% of each dataset is set aside for model validation and is used to tune hyperparameters. The remaining 80% is used for training.

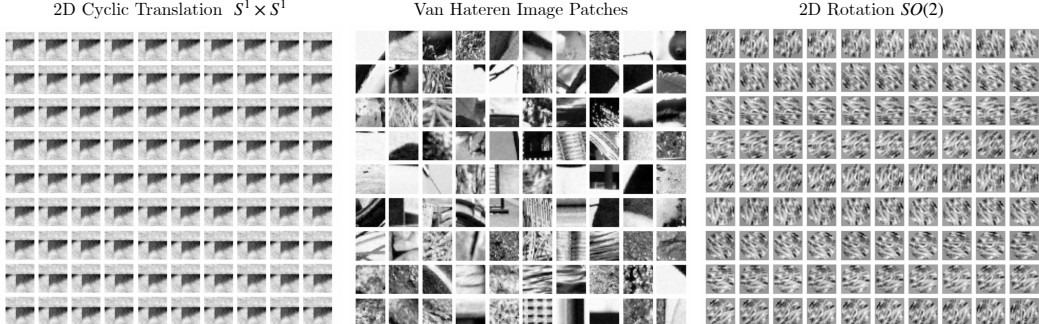

**Figure 8: Examples from the Cyclic 2D Translation and 2D Rotation Datsets.** Image patches are randomly sampled from the Van Hateren natural image database [35]. The orbit of each patch under 2D translation (Left) and rotation (Right) are generated.

## H.2 ROTATED MNIST

The Rotated MNIST dataset is generated by rotating each digit in the MNIST [36] training and test datasets by a random angle. This results in a training and test sets with the standard sizes of $60,000$ and $10,000$. A random $10\%$ of the training dataset is set aside for model validation and is used to tune hyperparameters. The remaining $90\%$ is used for training. Images are additionally downsized with interpolation to $16 \times 16$ pixels.

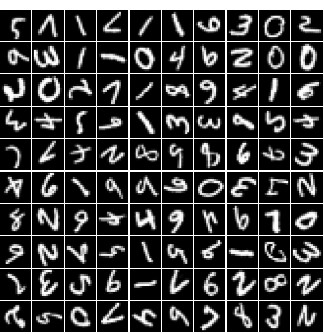

**Figure 9: Examples from the Rotated MNIST Dataset.**

## APPENDIX I  TRAINING PROCEDURES

All networks were implemented and trained in PyTorch [37]. We additionally made use of the open-source CplxModule library [38], which provides implementations of the various complex-valued operations and initializations used in this work.

### I.1 LEARNING FOURIER TRANSFORMS ON GROUPS

The weight matrix $W$ is a square complex-valued matrix of dimension $256 \times 256$, matching the dimension of the (raveled) $(16, 16)$ image patches in the dataset. Weights were initialized using the complex orthogonal weight initialization method proposed in [39]. Each parameter vector in $W$ was normalized to unit length after every gradient step. We used a batch sampler from the Pytorch Metric Learning library [40] to load batches with $M$ random examples per class, with $M = 10$ and a batch size of 100. Networks were trained with the Adam optimizer [41] until convergence, using an initial learning rate of $0.002$ and a cyclic learning rate scheduler [42], with $0.0001$ and $0.005$ as the lower and upper bounds, respectively, of the cycle.

### I.2 GROUP-INVARIANT CLASSIFICATION

The images in the Rotated MNIST dataset were down-scaled with interpolation to $16 \times 16$ pixels to match the trained model size. The dataset is passed through the Bispectral Network to yield a rotation-invariant representation of the images. The complex-valued output vector is reduced to 500 dimensions with PCA. The real and imaginary components of the output vectors are then horizontally concatenated before being passed to the classifier network.

The classifier model consists of a four-layer feedforward network with hidden layer dimension [128, 128, 64, 32], ReLU activation functions between each hidden layer and a softmax activation at the output that generates a distribution over the 10 digit classes. The MLP is trained with the cross-entropy loss on the labeled digits using the Adam optimizer with an initial learning rate of $0.002$, which is reduced by a factor of $0.95$ when the validation loss plateaus for 30 epochs, until a minimum learning rate of $0.00005$ is reached. Weights in the network are initialized using the Kaiming initialization [43] and biases are initialized to zero. The network is trained until convergence at 1500 epochs.The baseline Linear SVM model is trained with the L2-penalty, a squared-hinge loss, and a regularization parameter of $1.0$.

### I.3 COMPLETENESS

Inputs $\{\bar{x}_1, ..., \bar{x}_N\}$ are optimized to convergence using the Adam optimizer [41]. An initial learning rate of $0.1$ is used for all models, which is reduced by $0.1$ when the loss plateaus for 10 epochs.

### I.4 RECOVERING CAYLEY TABLES

Models were trained to convergence with the Adam optimizer [41] using cyclic triangular learning rate scheduler [42], a base learning rate og $10^{-5}$, a mximum learning rate of $0.001$, a and a step size up of $5$ epochs.

## APPENDIX J  LEARNING FOURIER TRANSFORMS ON GROUPS

We show the model's learned weights $W$ for each experiment: 2D Cyclic Translation and 2D Rotation. Each tile is the real part of one row of $W$ reshaped into a $16 \times 16$ image.

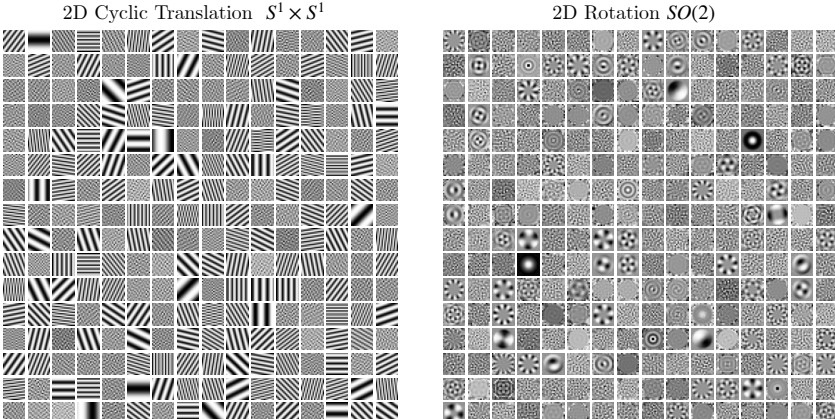

**Figure 10: Learned Filters**. Real components of weights learned in Bispectral Networks trained on the van Hateren Natural Images dataset under (left) Cyclic 2D Translation and (right) 2D Rotation.

## APPENDIX K   RECOVERING CAYLEY TABLES FROM IRREPS

Below we list the algorithms used to recover Cayley tables from a group's irreducible representations.

---

**Algorithm 1** GETCAYLEYFROMIRREPS

Compute the Cayley table from the (unordered) irreducible representations of a commutative group.

---

**procedure** GETCAYLEYFROMIRREPS($W$)
    **for** $i = 1, ..., n$ **do**
        **for** $j = 1, ..., n$ **do**
            $\rho \leftarrow W_i \odot W_j$
            $k^* \leftarrow \underset{k}{\mathrm{argmax}} |\langle \rho, W_k \rangle|$
            $\hat{C}(i, j) \leftarrow k^*$
        **end for**
    **end for**
    **return** $\hat{C}$
**end procedure**

---

**Algorithm 2** GETCAYLEYFROMGROUP

Compute the Cayley table of a group $G$.

---

**procedure** GETCAYLEYFROMGROUP($G$)
    **for** $g_1 \in G$ **do**
        **for** $g_2 \in G$ **do**
            $C(g_1, g_2) \leftarrow g_1 \cdot g_2$
        **end for**
    **end for**
    **return** $C$
**end procedure**

---

**Algorithm 3** ISISOMORPHIC

Check whether the learned group and the provided group are isomorphic by checking equality of their Cayley tables under permutation of group elements.
Let $S_n$ be the group of all permutations of $n$ elements.

---

**procedure** ISISOMORPHIC($G, W$)
    **for** $\pi \in S_n$ **do**
        $G_\pi \leftarrow \pi \circ G$       ▷ Permute the elements of $G$ according to the permutation element $\pi$
        $C_\pi \leftarrow$ GETCAYLEYFROMGROUP($G_\pi$)
        $\hat{C} \leftarrow$ GETCAYLEYFROMIRREPS($W$)
        **if** $C = \hat{C}$ **then**
            **return** True
        **end if**
    **end for**
    **return** False
**end procedure**

