# OpenReview forum: "Bispectral Neural Networks"
_ICLR.cc/2023/Conference — ICLR 2023 poster_

### Official Review · Reviewer_QqWn · 2022-10-21

**Confidence:** 3
**Correctness:** 3
**Technical Novelty And Significance:** 3
**Empirical Novelty And Significance:** 2
**Recommendation:** 6

**Clarity, Quality, Novelty And Reproducibility:**

In table 1, what are the FLOPs of each method? It would help to compare them in addition to the comparison of accuracy.

The paper mentions that it is not outperforming the compared methods but it has the benefit of learning the group from the data. However, it is not clear to me what is the real-world application of this; definitely I do not notice any related experiment.

In sec. 4.3 there is the claim that: “The goal of many adversarial attacks is to discover input points which are similar to a target sample in the model’s representation space, yet perceptually dissimilar to the true sample in the input space.”. Are there some citations for this?

Many of the most popular adversarial attacks are FGSM, PGD, C&W; in all of these cases the idea is the input image to be perceptually similar to the adversarial image. Are there any results on those benchmarks to demonstrate how the proposed network is more robust to those common attacks? That is, since the manuscript mentions that the method is “strongly robust to adversarial attacks”.

It’s not clear to me what is the role of the proposed bispectral block or the orbit separation loss in practice. Could the authors elaborate on their relationship and the sensitivity to the hyper-paremeter $\gamma$?

To continue on the previous line: are there any numerics on adversarial robustness when compared with other methods? At the moment only table 1 has numeric results in the main papers and most of the methods score above > 98%, which does not make it easy to appreciate the contributions of the proposed method.

Minor: Instead of “the context of vision” it should be the context of image/scene recognition in the introduction, since not all the vision tasks require the same symmetries.


**Strength And Weaknesses:**

 * [+] Including symmetries in neural networks is an interesting research direction.

 * [+] The writing is mostly clear, while I like the summary of the group notation in the appendix.

 * [+] The proposed block for the bispectrum is simple.

 * [-] The experiments are weak at the moment. To be specific, I do not find much the application of this in real-world data in the experimental section.

 * [-] The claim on the adversarial robustness seems exaggerated (see below).


**Summary Of The Paper:**

The idea is to learn inverariance to commutative groups. The groups are learned from data through the proposed block, which relies on the bispectrum idea. The proposed block (depicted in Fig. 2) resorts to a third-order polynomial that can be implemented on standard frameworks, as it requires an affine transformation of the input and a Hadamard product. In addition to this, an orbit separation loss is proposed.

**Summary Of The Review:**

The proposed bispectral block is new to me, and the idea of including invariances to groups in the network is also an interesting research direction. I would say one of the weaknesses in my mind is the experimental validation that is currently weak, given the empirical nature of this work.

_____________
After the rebuttal: See the discussion below, but the core idea is that some of the concerns have been addressed below.

---

> ### Author Response · Authors · 2022-11-19
> **Thank you and responses to questions**
>
> Dear Reviewer QqWn,
>
> Thank you for the close reading and detailed feedback. We respond to your questions in several comments below:

---

> > ### Author Response · Authors · 2022-11-19
> > **Adversarial Robustness**
> >
> > **Relation to common attacks: FGSM, PGD, C&W**
> >
> > The common attacks you mentioned highlight excessive *sensitivity* *to imperceptible noise.* The purpose of our adversarial robustness experiments is to test instead for *excessive invariance.* We base these experiments off of the paper “Excessive invariance causes adversarial vulnerability” [17]. Jacobsen, et al define an *Invariance-based Adversarial Example* as a point in the input space which yields identical neural activations and classification as a chosen target point, yet, is inequivalent with respect to a chosen oracle—e.g. perceptual equivalence [17, page 3]
> >
> > We chose this specific adversarial attack because it demonstrates the key property of our model: *complete invariance* (defined on page 3). Many common operations used to achieve invariance in deep learning architectures, such as max and avg, are *excessively invariant*, as they destroy signal structure along with the transformation. Our model, by contrast, is complete. This property arises directly from the theory and is borne out in experiment.
> >
> > We show empirically that the only input points that have the same neural activations in our model are points which are *related by a group action.* Performing the same experiment on the other models reveals that they lack this property: there exist points arbitrarily close in representation space to the target sample—classified as equivalent to the target—which perceptually bear no resemblance (Figure 5).
> >
> > We appreciate you raising this point as we realize we were not sufficiently clear about the motivation for the kind of attack we perform here, nor the kind of robustness we claim our model possesses. We have (1) introduced the terminology from reference [17] in 4.3, (2) updated the abstract and discussion sections to clarify that our results demonstrate strong *invariance-based* adversarial robustness.
> >
> >
> > **Quantification of Robustness**
> >
> > To quantify the susceptibility of the three models to invariance-based attacks, we ran an extension of the experiment presented in Section 4.3. In this new experiment, we selected 70 random targets from the Rotated MNIST dataset for each model. Starting from random uniform initializations, we drove inputs towards the model representations of the targets.
> >
> > Inputs with sufficiently similar representations can be thought of as "metamers" for the model (stimuli that are physically non-equal but are perceived to be equivalent, see [Metamers of the Ventral Stream](https://www.nature.com/articles/nn.2889)). Model metamers that are not also metamers for the human visual system, up to group action (the oracle) are counted as successful invariance-based attacks.
> >
> > The figure linked below shows the optimized metamers for E2CNN, Augerino, and Bispectral Networks.
> >
> > [FIGURE - invariance-adversaries.png](https://postimg.cc/B8gH7rHt)
> >
> > For E2CNN, all of the optimized model metamers are unrelated to the target images, revealing excessive invariance in the model. Interestingly, for Augerino, more structured features emerge that appear to resemble components of the target digits. However, they are still far from perceptually equivalent images, and include considerable noise and other artifacts. For the Bispectral Network, by contrast, all of the optimized metamers are simply rotated versions of the target. Thus, BNNs show strong and consistent robustness to invariance-based adversarial attacks.

---

> > > ### Comment · Reviewer_QqWn · 2022-11-21
> > > **Follow-up on the responses**
> > >
> > > Dear authors,
> > >
> > > I am thankful for the detailed answers. Even though I still believe the experiments are weak (as raised by another reviewer as well), I would be willing to increase the score to 6 if the FLOPS are reported (as mentioned in the original review) and are reasonable. In addition, I strongly recommend to include the answer on the [motivation](https://openreview.net/forum?id=xnsg4pfKb7&noteId=4J5modQpQI) to the paper, even to the appendix if there is no space in the main paper.

---

> > > > ### Author Response · Authors · 2022-11-22
> > > > **FLOPS and computational efficiency**
> > > >
> > > > Dear Review QqWn,
> > > >
> > > > Thank you for taking our responses into consideration and we appreciate your willingness to adjust your score. We agree that the paper would benefit from the incorporation of our response on the motivation / real-world applications into the paper and the inclusion of computational efficiency calculations in the main table. We would be happy to make these modifications in the final paper.
> > > >
> > > > Neither of the papers we compare to (E2CNN or Augerino) report FLOPS for their models, so we are working on these calculations now. There is a slight hangup on acquiring these numbers, because both models involve many custom modules and operations, which throw errors for FLOP calculator libraries such as [fvcore](https://github.com/facebookresearch/fvcore/blob/main/docs/flop_count.md).
> > > >
> > > > If we are unable to get an accurate FLOP count for all three models, we are wondering if another measure of computational cost might be acceptable in its place? For example, we could record the time it takes to complete one forward pass, for the same sized inputs on the same hardware for all three.
> > > >
> > > > In addition, we can provide the computational complexity in big O notation for our model. Again, neither E2CNN or Augerino report this metric. In this case, we would only be able to report ours.
> > > >
> > > > Let us know what you think, or if there are other metrics you think would be more meaningful.

---

> > > > > ### Comment · Reviewer_QqWn · 2022-11-23
> > > > > **Computational efficiency**
> > > > >
> > > > > I assume the big O notation and practical runtime would be ok.

---

> > > > > ### Author Response · Authors · 2022-11-23
> > > > > **Computational efficiency**
> > > > >
> > > > > Dear Reviewer QqWn,
> > > > >
> > > > > Following up here with calculations:
> > > > >
> > > > > We recorded the average time of a forward pass for the models we compare to, with inputs of size (16, 16), batch size 2, averaged over 100 iterations, on an NVIDIA GeForce GTX 1080 Ti. Results are below:
> > > > >
> > > > > `E2CNN: 0.0107 seconds`
> > > > >
> > > > > `Augerino: 0.0150 seconds`
> > > > >
> > > > > `BNNs: 0.0130 seconds`
> > > > >
> > > > >
> > > > > With this analysis, there are slight but not particularly substantial time differences between the three methods.
> > > > >
> > > > > However, we will note that our model permits significant reductions in time complexity through the use of the symmetry-reduced form presented in Appendix H. Theoretically, this reduces the time complexity of the bispectral layer from $\mathcal{O}(n^2)$ to $\mathcal{O}(n)$. Empirically, for this analysis, the cost of a forward pass is reduced by approximately one order of magnitude:
> > > > >
> > > > > `BNNs Symmetry-Reduced (Appendix H): 0.0029`
> > > > >
> > > > > The results presented in the paper here did not use the symmetry-reduced form, for clarity of presentation, but we anticipate making use of this reduction as we scale the method in future work. Further efficiency gains can be achieved by making the model "convolutional" rather than fully-connected.

---

> > > > > > ### Comment · Reviewer_QqWn · 2022-11-23
> > > > > > **Raising the score to 6**
> > > > > >
> > > > > > Dear authors,
> > > > > >
> > > > > > I am thankful for your prompt response. Assuming this table will be included in the final paper (along with a discussion on the computational cost), I increase my score to 6.

---

> > > > > > > ### Author Response · Authors · 2022-11-23
> > > > > > > **Computational Efficiency**
> > > > > > >
> > > > > > > Thank you very much. We confirm that we will add the analysis and discussion of the computational cost to the final paper.

---

> > ### Author Response · Authors · 2022-11-19
> > **The Loss Function**
> >
> > *Could the authors elaborate on their relationship and the sensitivity to the hyper-paremeter γ?*
> >
> > Thanks for your question; hopefully this will clarify the components of the loss function. The loss is similar in spirit to a contrastive loss. The first term is the “orbit collapse term” because it seeks to collapse each orbit to a single point. It does this by bringing the positive pairs closer together in the representation space—this encourages the network to learn a map which is invariant to transformations. The second term is an “information-preservation term” which pushes the weight matrix W towards orthonormality, to ensure that the layer does not trivially collapse all points to one point. It does this by encouraging the matrix W to be invertible through multiplication with its own conjugate transpose.
> >
> > Empirically, we find that the information-preservation term is rapidly reduced to near-zero in the first few epochs of training, while the orbit collapse term decreases more slowly. Why does this happen? Intuitively, it is easier to push a weight matrix towards orthonormality than to find an invariant map. We have found that  training has not been very sensitive to the magnitude of the penalty term in the second term of the loss. We do notice that if the gamma parameter is too small (< 0.1), then the linear layer weights may no longer converge to an orthonormal matrix. Above that level, however, we have seen comparable results for different initializations.  In these experiments, we chose the value 0.56.

---

> > ### Author Response · Authors · 2022-11-19
> > **Real-World Applications**
> >
> > *"The paper mentions that it is not outperforming the compared methods but it has the benefit of learning the group from the data. However, it is not clear to me what is the real-world application of this; definitely I do not notice any related experiment."*
> >
> > Regarding real-world applications, we would like to make two points.
> >
> > First, the problem of *learning a group from data* is one that has been appreciated in many prior works: for example, Sejnowski, Kienker, Hinton (1986), Rao & Ruderman (1999), Dickstein, Wang, Olshausen (2010), Cohen and Welling (2014).
> >
> > There are several reasons why this is an interesting problem:
> >
> > - The ability to learn (on the fly) the symmetries in data is one of the hallmarks of intelligence, as exemplified by classic “IQ” tests such as Raven’s matrices.
> > - A model of the symmetry structure in data allows for few-shot learning and generalization.
> > - Although many symmetries present in natural data can be specified a priori (e.g. translation and rotation), biology is able to adapt to novel unseen symmetry groups. This is still a challenging problem for AI.
> > - Moreover, group structure often acts on invisible *latent variables* in data, resulting in complex nonlinear transformations in the data space. The common existing paradigm for group-equivariant deep learning involves hand-specifying a linear group action (convolution) on the data domain. This will not work for such cases; flexible methods that allow for group learning are more appropriate.
> >
> > Prior works for group learning (mentioned above) had difficulty scaling because they depended on computationally expensive machinery (the exponential map from Lie theory and EM inference). In our work, we offer a novel path for group learning that uses algebraic machinery (irreducible representations and the bispectrum) rather than the tools of Lie theory. A key advantage of our approach is the simplicity of the computation, as it is completely feedforward. The exponential map approach has been explored for several decades; we expect that exploration of the path we lay here will yield further advances, given its key advantages.
> >
> > Second, we acknowledge that the datasets we use in the paper are relatively simple. Indeed, we do not propose the network as a standalone solution to large-scale complex vision tasks. Instead, we view our simple 2-layer network as a *module* with rigorously defined desirable properties—a useful computational primitive which can be incorporated in various ways into a wide variety of models for a range of signal modalities. In this paper, we aimed to establish and extensively characterize properties of the learned bispectrum in relation to the analytical object on carefully-chosen, simple datasets. These simple datasets allow us a standard way to compare our method’s performance and possible limitations to the other methods in our subfield (e.g. RotMNIST for E2CNN, Augerino, Cohen and Welling 2014).
> >
> > Our work is the first to demonstrate two key achievements (1) that a bispectrum and group can be simultaneously learned solely from the symmetries in data alone, (2) that a group Cayley table, the most explicit representation of group structure, can be learned from data. In the context of equivariant deep learning, these are significant advancements that required substantial theoretical work to accomplish. Given limited space, we believe that it is most important to first thoroughly establish these foundations. We aim to tackle the problem of incorporating this module into larger systems for more complex machine learning tasks in future work.
> >
> > Nonetheless, there exist many real-world problems at a similar scale of complexity as the experiments performed here -- For example, constructing invariant representations of CryoEM images of proteins, solving small graph isomorphism problems for molecular or biological networks, and other signal processing applications to which the analytical bispectrum has been applied. We are looking towards these applications in future work.

---

### Official Review · Reviewer_pa8G · 2022-10-24

**Confidence:** 2
**Correctness:** 4
**Technical Novelty And Significance:** 3
**Empirical Novelty And Significance:** 3
**Recommendation:** 8

**Clarity, Quality, Novelty And Reproducibility:**

This paper presents a novel approach for learning groups from data in an efficient, interpretable, and fully feed-forward model that requires no prior knowledge of the group, computation of the exponential map, or bayesian inference.

**Strength And Weaknesses:**

strengths:
1. This paper has comprehensive and rigorous mathematics proof.
2. The network is simple yet effective, and the competitive results show the effectiveness.
3. This paper has demonstrated that the completeness property endows these networks with strong adversarial robustness.

**Summary Of The Paper:**

This paper proposes a neural network architecture, Bispectral Neural Networks(BNNs), which aims to learn the group-invariant representations of the actions of compact commutative groups. This simple architecture is composed of two layers: a single learnable linear layer, followed by a fixed collection of triple products computed from the output of the previous layer. Then, some comparison experiments are conducted to show the effectiveness of the proposed method.

**Summary Of The Review:**

From the manuscript, the authors propose BNNs, which could simultaneously learn groups, their irreducible representations, and corresponding complete invariant maps purely from symmetries implicit in data. I think it is good.

---

> ### Author Response · Authors · 2022-11-19
> **Thank you and request for clarification**
>
> Dear Reviewer Pa8G,
>
> We appreciate your time spent reading our paper and thank you for your review.
>
> We noticed that you gave the paper a correctness score of 3, meaning that “Some of the paper’s claims have minor issues. A few statements are not well-supported, or require small changes to be made correct.” We’d like to improve the paper as much as possible, for ourselves and for other readers. In that spirit, could you share some of the issues with the claims which could be made stronger or more clear? This would be very helpful.

---

> > ### Comment · Reviewer_pa8G · 2022-11-23
> > **Thanks and clarification**
> >
> > Dear authors:
> >
> > Thanks for your replies.
> > After re-reading the paper and comments (not only limited to my comments), I have corrected my review scores.

---

> > > ### Author Response · Authors · 2022-11-23
> > > **Thank you**
> > >
> > > We appreciate the additional time you've put into reading and reviewing the paper.

---

### Official Review · Reviewer_1MdX · 2022-10-26

**Confidence:** 3
**Correctness:** 4
**Technical Novelty And Significance:** 4
**Empirical Novelty And Significance:** 3
**Recommendation:** 8

**Clarity, Quality, Novelty And Reproducibility:**

The quality of the paper is exceptionally high. As someone relatively new to the field of equivariances in machine learning and entirely new to group representation theory, I found the paper extraordinarily pedagogical. Appendix A is very helpful and section 2 is well-written and easy to understand despite the abstract topic. The method itself appears to be novel (I wasn't able to find anything similar) and the ideas and concepts could have a significant impact on the fields of invariances and equivariances in machine learning. The authors provide the code to reproduce all experiments and figures, though I did not try to run this. In general, the paper should be of significant interest to the ICLR community.


**Questions for the authors**

1. For the loss, did you experiment with other norms? For images, in particular, the $L_2$ norm is not very informative, but I cannot intuitively see if this could cause problems.
2. As mentioned under "weaknesses", it is unclear to me how well the method scales. I don't expect any new experiments, but do you have a feeling for how large inputs the method can handle?

**Strength And Weaknesses:**

### Strengths
Overall, I find this paper very strong. While the bispectrum is extensively used in other fields, it does not seem to be the case in machine learning (as far as my searching abilities go). Simply introducing this incredibly useful concept to the machine learning community has value in itself, and then further parametrising it in terms of a neural network, which, given a novel loss function, learns the group of transformations directly from data, makes this paper very strong. The simplicity and elegance of the proposed method are intriguing, and I could see the ideas introduced here forming the basis of many future papers on invariant and equivariant models.

In summary, the paper
- presents an incredibly useful concept (the bispectrum),
- introduces a novel and interesting loss function, and
- demonstrates how to learn irreducible representations of the transformation group using a neural network.


### Weaknesses
The paper appears to be mostly a proof-of-concept. The datasets used in the experiments are quite simple, which the authors acknowledge themselves. It is also unclear to me if the method is usable in practice in terms of the computations that are required to compute the bispectrum. Since the complex weight matrix needs to be of size $n \times n$, $n$ being the number of input dimensions, does the method scale to inputs of even moderately large dimensionality? A study of this, e.g., in terms of the training time and memory cost for increasing image sizes, would have been interesting. The authors say that they are currently working on a localised version of the model, such that it can be used in a convolutional neural network, which would address this issue. Still (and without knowing exactly how much work this entails), one might have expected such an extension to be included in the current paper.

I see these weaknesses as minor, though, given the significance of the model and the ideas and concepts the paper introduces.

In summary, I see the main weaknesses as being that
- the model is only tested on simple datasets, and that
- it is unclear if the model scale to inputs of moderately large dimensionality.

**Summary Of The Paper:**

The paper presents the Bispectral Neural Network (BNN), a network primitive which learns to be invariant to transformations present in the data. The model is defined in terms of two main components. Firstly, the model treats the irreducible representations of a group of transformations as weights in the network, thus essentially learning a Fourier transform on the group, which is equivariant to the group action. The Fourier representation of the input is then mapped to the bispectrum, which is invariant to the transformation but retains the information needed to uniquely restore the image (up to the transformation), by the network. Secondly, a novel loss function encourages the bispectrum for inputs of the same class to be identical, effectively making the trained network invariant to the transformation. A crucial point here is that the group of transformations need not be defined a priori; the network discovers the transformations automatically.

The authors test the model in four different experiments, demonstrating that it can construct the transformation group from data alone, that the learnt irreducible representations are transferable to other datasets, and that the model is robust to adversarial attacks.

**Summary Of The Review:**

The paper presents a novel method for learning representations of data that are invariant to transformations of the data. Not only is the method novel, the ideas and concepts presented in the paper are intriguing and should be of significant interest to the ICLR community. The experiments are somewhat weak, though, and the paper seems to be mostly a proof-of-concept, yet the strengths far outweigh these weaknesses. I view the paper as a very significant and impactful contribution to the fields of invariance and equivariance in machine learning.

---

> ### Author Response · Authors · 2022-11-19
> **Thank you and responses to questions**
>
> Dear Reviewer 1MdX,
>
> Thank you for your careful read of our paper. We greatly appreciate your thorough analysis of its merits, potential next steps, and avenues for improvement, and respond to these points in several comments below:

---

> > ### Author Response · Authors · 2022-11-19
> > **Scaling Up**
> >
> > Regarding scaling:
> >
> > The simplest form of the model, which we present in the paper for clarity, has $\frac{n^2}{2} - n$ output neurons. However, there are several symmetries in the bispectrum (Appendix G) that can be exploited to reduce the dimension to order $n \times k$, without loss of information, with $k$ the dimension of the group. In practice, $k$ is typically small for many groups of interest (i.e. $k=1$ for 2D rotation and $k=2$ for 2D translation). We introduce this reduced form, which scales linearly with the size of the patch, in a new Appendix H “Symmetry Reduction of the Model.”
> >
> > Another avenue for scaling to larger images comes about through applying the bispectral module to local patches instead. We are currently developing a localized, “convolutional” model that uses the primitive we propose in this paper to learn bispectra for each local receptive field at each stage in the hierarchy. We are currently exploring methods of stacking these layers in a deep network to examine its performance on more complex datasets.
> >
> > The module we propose in this paper incorporates several components unconventional in deep learning: complex-valued parameters, third order interactions, Fourier analysis and bispectra on groups. We thus thought it most sensible to first introduce the module and its theoretical underpinnings on its own -- demonstrating its performance and properties on simple tasks -- before building on these results and incorporating the module into more complex systems.

---

> > ### Author Response · Authors · 2022-11-19
> > **L2 Norm in the Loss**
> >
> > Regarding the $L_2$ Norm:
> >
> > Indeed, for images the $L_2$ norm computed in pixel space is not very informative. However, in our case we are using the $L_2$ norm on the output of the network, i.e. in “bispectrum space,” which is a complex-valued space defined over pairs of frequencies, with values indicating the strength of their couplings. The bispectrum canonicalizes inputs by quotienting out group actions. Thus, intuitively, distance in bispectrum space captures the distance between canonical exemplars. We chose the $L_2$ norm because it behaves well in the complex field, but we have not experimented much with other norms. However, it is certainly worth examining, and we appreciate the suggestion.

---

> > ### Author Response · Authors · 2022-11-19
> > **Simple Datasets**
> >
> > Responding to the following point: *"The paper appears to be mostly a proof-of-concept. The datasets used in the experiments are quite simple, which the authors acknowledge themselves."*
> >
> > [Note: this comment is also copied below as part of the response to reviewer QqWn - Real-World Applications]
> >
> > Our primary goal in this work was to establish and thoroughly characterize the link between the group-invariant bispectrum and learnable invariant maps for object recognition through our mathematical construction, experiments, and analysis.
> >
> > We acknowledge that the datasets we use in the paper are relatively simple. Indeed, we do not propose the network as a standalone solution to large-scale complex vision tasks. Instead, we view our simple 2-layer network as a module with rigorously defined desirable properties—a useful computational primitive which can be incorporated in various ways into a wide variety of models for a range of signal modalities. In this paper, we aimed to establish and extensively characterize properties of the learned bispectrum in relation to the analytical object on carefully-chosen, simple datasets. These simple datasets allow us a standard way to compare our method’s performance and possible limitations to the other methods in our subfield (e.g. RotMNIST for E2CNN, Augerino, Cohen and Welling 2014).
> >
> > Our work is the first to demonstrate two key achievements (1) that a bispectrum and group can be simultaneously learned solely from the symmetries in data alone, (2) that a group Cayley table, the most explicit representation of group structure, can be learned from data. In the context of equivariant deep learning, these are significant advancements that required substantial theoretical work to accomplish. Given limited space, we believe that it is most important to first thoroughly establish these foundations. We aim to tackle the problem of incorporating this module into larger systems for more complex machine learning tasks in future work.

---

> > ### Comment · Reviewer_1MdX · 2022-11-24
> > **Thank you for your responses. A couple of follow-up questions.**
> >
> > Thank you for responding to my questions, and apologies for my late reply.
> >
> > Regarding the $L_2$ norm, you make a good point for the norm in the bispectrum space. However, the second term in Eq. (10) is the $L_2$ norm of the difference between the original image and its reconstruction, is it not? Will the norm not be a problem here?
> >
> > Thank you for the addition of appendix H; it is very much appreciated. It is fascinating how you can make use of the bispectrum symmetries to reduce the computational complexity, and the timings you provide for reviewer QqWn are striking. Thinking of $k$ as a hyperparameter when the symmetries are unknown is hugely appealing from a practitioner's point of view.
> >
> > Just to see if I understand this correctly, the largest you would ever set $k$ to would be $k = \frac{n^2}{12(n+1)} \approx \frac{n}{12}$, due to the inherent symmetries of the bispectrum? This could then be further reduced if you have additional knowledge of the symmetries in the data (or are willing to tune the parameter empirically).

---

### Comment · Area_Chair_1err · 2022-11-22
**Please respond as soon as possible if you still have questions on the paper.**

Please respond as soon as possible if you still have questions on the paper.

---

> ### Comment · Area_Chair_1err · 2022-12-05
> **Zoom Meeting**
>
> For all reviewers, which have not responded to the authors, I will have to ask you to meet via Zoom. If you want to avoid such an additional step, please respond by Dec. 5.

---

### Decision · Program_Chairs · 2023-01-20

**Decision:**

Accept: poster

**Justification For Why Not Higher Score:**

The result in this paper is not so significant.

**Justification For Why Not Lower Score:**

NA

**Metareview: Summary, Strengths And Weaknesses:**

This paper presents a new neural network called Bispectral Neural Networks (BNNs) that is designed to learn group-invariant representations of the actions of compact commutative groups. The architecture of BNNs consists of two layers: a single learnable linear layer followed by a fixed collection of triple products based on the output of the previous layer. The authors conduct experiments to demonstrate the effectiveness of BNNs.

The reviewers are positive about the paper. Though minor concerns remain, they believe the author further improve the paper in the camera ready version.

**Note From Pc:**

if the above contains the word "oral" or "spotlight" please see: "oral" presentation means -> notable-top-5% and "spotlight" means -> notable-top-25%. As stated in our emails, we are disassociating presentation type from AC recommendations

**Summary Of Ac-Reviewer Meeting:**

NA